# The mechanism of NF-κB-TERT feedback regulation of granulosa cell apoptosis in PCOS rats

Haoxuan Xue[1], Zecheng Hu[2], Shun Liu[1], Shun Zhang[3], Wenqin Yang[1], Jiasi Li[1], Chulin Yan[1], Jiaming Zhang[1]*, Jing Zhang[1]*, Xiaocan Lei[1]*

1 The First Affiliated Hospital, Gynecology & Obstetrics and Reproductive Medical Center, Clinical Anatomy and Reproductive Medicine Application Institute, School of Basic Medical Sciences, Hengyang Medical School, University of South China, Hengyang, Hunan, China, 2 Department of Breast and Thyroid Surgery, The First Affiliated Hospital, Hengyang Medical School, University of South China, Hengyang, Hunan, China, 3 Department of Reproductive Medical Center, The Affiliated Hospital of Guilin Medical University, Guilin, China

☉ These authors contributed equally to this work.
* zhangjia_ming123@163.com (JZ); 41314619@qq.com (JZ); liulincun@163.com (XL)

**Data Availability Statement:** All relevant data are within the paper and its Supporting information files.

**Funding:** This work was supported by the Scientific Research Foundation of Hunan Provincial

## Abstract

Patients with Polycystic ovary syndrome (PCOS) have chronic low-grade ovarian inflammation. Inflammation can cause telomere dysfunction, and telomere and telomerase complex are also involved in regulating inflammation. However, the specific mechanisms of inflammatory signaling feedback and telomere-telomerase mutual regulation remain to be discovered. This study elucidates the role of Nuclear factor kappa-B (NF-κB)-Telomerase reverse transcriptase (TERT) feedback in PCOS granulosa cell apoptosis. Using letrozole and a high-fat diet, a PCOS rat model was established, along with a Lipopolysaccharide (LPS)-treated KGN cell inflammation model was established. NF-κB and TERT inhibitors (BAY 11–7082 and BIBR1532) were then administered to LPS-induced KGN cells. PCOS rats displayed disrupted estrous cycles, increased weight, elevated serum testosterone, cystic follicles, granulosa cell layer thinning, and reduced corpora lutea count ($P$ are all less than 0.05). In PCOS rat ovaries, NF-κB, Interleukin-6 (IL-6), Tumor Necrosis Factor α (TNF-α), TERT, Bax, and Caspase-3 exhibited notable upregulation, while Bcl-2 decreased, with telomere elongation ($P$ are all less than 0.05). There were significant correlations among NF-κB-related inflammatory factors, TERT and apoptotic factors, and they were positively correlated with Bax and Caspase-3, and negatively correlated with Bcl-2 ($P$ are all less than 0.05). LPS-treated KGN cells demonstrated increased expression of inflammatory and pro-apoptotic factors, later restored post-treatment with NF-κB and TERT inhibitors ($P$ are all less than 0.05). In conclusion, TERT may induce granulosa cell apoptosis by participating in the regulation of the NF-κB signaling pathway, thereby mediating the chronic inflammatory response of PCOS through downstream inflammatory factors IL-6 and TNF-α.

Education Department [20A434], The Natural Science Foundation of Guangxi in China [2020GXNSFAA297097] to SZ, The Natural Science Foundation of Hunan Province, China [2020JJ5511] to JiaZ and [2024JJ9373] to JinZ, Scientific Research Foundation of Hunan Provincial Education Department [22B0406] to JiaZ, Health Research Foundation of Hunan Provincial Health Commission Scientific Research [W20243058] to JiaZ, Foundation of Hengyang City science and technology innovation plan [202330046470] to JiaZ, and the Hunan Province Innovation and Entrepreneurship Training Program for College Students [S202310555229, S202310555214, D202305162026050688].

**Competing interests:** The authors have declared that no competing interests exist.

## Introduction

PCOS is a common and complex endocrine, metabolic and psychological disorder affecting 5–18% of women in reproductive age [1, 2]. The Rotterdam criteria (2003) are commonly employed for diagnosing PCOS, requiring the presence of at least two of the following three features: clinical and/or biochemical hyperandrogenism, irregular ovulation or anovulation, and polycystic ovarian morphology (PCOM) [1, 3]. The cause of PCOS is multifaceted, including strong epigenetic and genetic influences, accompanied by hyperandrogenism, chronic inflammation, obesity, insulin resistance and abnormal lipid metabolism [4–6]. Research indicates that PCOS is characterized by a proinflammatory state, with emerging evidence supporting the presence of chronic low-grade ovarian inflammation in affected women [7, 8]. Patients with PCOS have significantly increased levels of proinflammatory factors, such as TNF-α, IL-6 and so on [9–11]. The pathway used to control TNF-α and IL-6 is through the Regulation of NF-κB [12]. What's more, It has been shown that the NF-κB plays an important role in the inflammatory state of patients with PCOS [7].

Telomeres are specific structures located at the ends of linear chromosomes in eukaryotes. The maintenance of telomere length is essential for the normal structure and function of telomeres and the integrity and stability of chromosome ends [13]. Each cell division shortens the length of telomere 5'-TTAGG-3' sequence, which eventually leads to apoptosis or senescence [14]. Telomerase is a ribonucleoprotein complex comprising TERT, telomerase RNA component (TERC), and telomerase-associated proteins (TEP) [15]. TERC was used as a template for telomerase. TERT acts as a catalytic subunit to reverse transcribe telomeric DNA and maintain telomere length [16]. The expression level of TERT is the main determinant of telomerase activity [17]. The initiation of telomerase is a prerequisite for maintaining telomere length and genome integrity and achieving cell immortalization and malignant transformation [18]. Most human somatic cells do not show telomerase activity, and a few cells with vigorous division, such as stem cells [19], tumor cells [18], and germ cells [20], express telomerase activity to varying degrees. The study included women aged 20 to 30 years, comprising 40 PCOS patients and 35 healthy controls for comparison. It was found that the telomere length of granulosa cells in PCOS group was significantly longer than that in Control group, while the telomere length of peripheral blood leukocytes was not different from that in Control group [21]. Another study showed that 75 PCOS patients and 81 non-PCOS women were included in the study, and it was found that there was no significant difference in leukocyte telomere length between the Control group and PCOS patients. Interestingly, however, when comparing the telomere length of granulosa cells between Controls and PCOS patients, the telomere length of PCOS patients was significantly longer. The p values remained significant after adjusting for age and body mass index [22]. Pedroso et al. proposed that the maintenance of telomere length in cumulus cells of PCOS patients might be attributed to elevated telomerase TERT activity [23].

It has been proved that telomerase can regulate NF-κB signaling pathways, resulting in the stimulation of TNF-α and IL-6 [24], and telomerase activity in cumulus cells was higher in the PCOS group [23]. TERT is a component of telomerase, and it interacts with NF-κB [25]. With Given this, it is crucial to ascertain whether there is an association between NF-κB-related chronic low-grade inflammation and TERT expression in PCOS ovaries. Recent studies have focused on the function of granulosa cells (GCs), which play an important role in folliculogenesis and follicle maturation [26]. Apoptosis is a form of programmed cell death, with the anti-apoptotic protein Bcl-2 and the pro-apoptotic protein Bax being the most representative. An imbalance in the interaction between Bcl-2 and Bax is a primary cause of apoptosis [27]. The Caspase family also plays a significant role in the molecular mechanisms that induce apoptosis,

serving as the final common pathway for various apoptotic signaling routes [28]. Apoptosis has been observed in GCs of PCOS patients [29]. Bax and Caspase-3 and Bcl-2 play a crucial role in regulating the growth and apoptosis of GCs during follicular development [27, 29, 30]. And chronic inflammation can cause continuous oxidative stress, leading to cell apoptosis [31]. Moreover, it has been reported that chronic low-grade inflammation in PCOS eventually leads to granulosa cell apoptosis [32]. Studies have confirmed that the chronic inflammatory response in PCOS patients can affect the changes in telomere length and telomerase activity [33]. Because the expression of TERT directly reflects the level of telomerase activity [13], and the activation of NF-κB signaling pathway can cause the release of TNF-α and IL-6. Hence, TERT may mediate chronic inflammatory response and induce apoptosis of granulosa cells by participating in the regulation of NF-κB signaling pathway. Therefore, this study will investigate the interaction between inflammation and telomeres and how this feedback influences granulosa cell apoptosis, ultimately leading to impaired follicular development in PCOS rats.

## Materials & methods

### Animals and experiment protocol

Female Sprague Dawley (SD) rats (n = 12, 5 weeks old, 172 ± 8g) were obtained from Hunan SJA Laboratory Animal Co.Ltd (Changsha, China). The rats were reared at 25°C, 12 h light/dark cycle, and enough food and water were provided for free access. After one week of adaptive feeding, the rats were randomly divided into control group (n = 6) and PCOS group (n = 6). The control group were supplied with standard pellets of feed for 30 days; the PCOS group were fed with a high-fat diet (HFD, consisting of 61.5% standard food, 12% lard, 5% sucrose, 5% milk powder, 5% peanut, 10% egg, 1% sesame oil, 0.5% salt) and letrozole (1 mg/kg/day, dissolved in 1% carboxymethylcellulose [CMC] for 30 days to establish the PCOS model [34]. All rats were anesthetized with an intraperitoneal injection of 20% urethane (0.6 mL 100 g$^{-1}$) before euthanasia. Body weight was determined at the end of modeling. Blood samples were collected after anesthetization. Ovaries were isolated, weighed, and the left ovaries of rats from each group was fixed in 4% paraformaldehyde overnight before being embedded in paraffin. The right ovaries of rats from each group were stored at −80°C for molecular analysis. All animal experiments were performed in accordance with Regulations on the Administration of Laboratory Animals of University of South China (Revised in 2017) and approved by the Animal Ethics Committee of University of South China (No. SYXK2020-0002).

### Vaginal smears and estrous cycle determination

Vaginal smears were taken daily between 11:00 am and 12:00 for 10 days before the end of modeling [35]. The estrous cycle stage was determined by observation of the vaginal smears using a light microscope (Zeiss,Germany). The stages of the estrous cycle were determined by identifying predominant cell types of cells present in rat vaginal smears [36–39]: nucleated epithelial cells (proestrus); cornified cells (estrus); and mixed cells (nucleated, cornified, leucocytes) (metestrus); leucocytes (diestrus).

### Serum hormone measurement

All rats were anesthetized with 20% urethane,and the whole blood sample was collected from the abdominal aorta of rats immediately. The serum was further isolated by centrifuging at 2000 rpm for 10 min at 4°C, and stored at −80°C for further determination. The levels of

serum T, follicle stimulatinghormone (FSH) and luteinizing hormone (LH) were determined with radioimmunoassay (Beijing North Institute of Biotechnology Co., Ltd., Beijing, China).

## Hematoxylin-eosin (H&E) staining

The left ovaries of rats from each group was fixed in 4% paraformaldehyde overnight before being embedded in paraffin, and then were sequentially sectioned with 5 μM- slices for hematoxylin and eosin (H&E) staining in order to examine the pathological structural alterations of the ovary.

## Real-Time Quantitative PCR (RT-qPCR) analysis

Total RNA from ovary tissues and cells was extracted using TRIZOL reagent (15596026, Thermo Fisher Scientific, Shanghai, China) and cDNA was synthesized with TransScript One-Step gDNA Removal and cDNA Synthesis SuperMix Kit (TRAN, Beijing, China). Real-time PCR analyses were performed with the 2× Universal SYBR Green Fast qPCR Mix Kit (RK21203, ABclonal) and Applied Biosystems QuantStudio 3 (Thermo Fisher Scientific). GAPDH was used as the reference and The critical threshold cycle (Ct) [38] value was determined for each reaction, which was transformed into relative quantification data using the $2^{-\Delta\Delta Ct}$ method. The experimental primers were Human IL-6 (HQP009670, GeneCopoeia), Human TERT (HQP018018, GeneCopoeia) and Human TNF-α (HQP018141, GeneCopoeia). The primer sequences used for amplification are shown in **Table 1**.

**Table 1. Primer sequences used for the qRT-PCR analysis.**

| Gene ID | | Sequence | GeneBank Accession NO. |
|---|---|---|---|
| Rat NF-κB | Forward | AGCCACTGCCTTCCCTACTTC | NM_001276711.2 |
| | Reverse | GGTCCTTAGCCCACTCCTTCTG | |
| Rat TNF-α | Forward | GTCCCAACAAGGAGGAGAAGT | NM_012675.3 |
| | Reverse | CTGGTATGAAATGGCAAATCG | |
| Rat TERT | Forward | AGTGGTGAACTTCCCTGTGG | NM_053423.2 |
| | Reverse | CAACCGCAAGACTGACAAGA | |
| Rat Bax | Forward | GAGACACCTGAGCTGACCTT | XM_032913059 |
| | Reverse | TCCATGTTGTTGTCCAGTTC | |
| Rat Bcl-2 | Forward | AGTACCTGAACCGGCATCT | NM_016993 |
| | Reverse | TCTTCAGAGACAGCCAGGA | |
| Rat Caspase-3 | Forward | CCGGTTACTATTCCTGGAGA | XM_006253130 |
| | Reverse | TAACACGAGTGAGGATGTGC | |
| Rat GAPDH | Forward | TACACTGAGGACCAGGTTG | XM_032916238 |
| | Reverse | CCCTGTTGCTGTAGCCATA | |
| Homo NF-κB | Forward | CTGCCGGGATGGCTTCTAT | NM_001165412.2 |
| | Reverse | CCGCTTCTTCACACACTGGAT | |
| Homo Bax | Forward | GCGACTGATGTCCCTGTCT | NM_001291430 |
| | Reverse | TGAGTGAGGCGGTGAGC | |
| Homo Bcl-2 | Forward | CCCTGTGGATGACTGAGTACC | NM_000633 |
| | Reverse | AGACAGCCAGGAGAAATCAAA | |
| Homo Caspase-3 | Forward | CAGTGATGCTGTGCTATGAAT | NM_001354783 |
| | Reverse | CAGATGCCTAAGTTCTTCCAC | |
| Homo GAPDH | Forward | GAGTCCACTGGCGTCTTCAC | NM_001357943.2 |
| | Reverse | GAGGCATTGCTGATGATCTTGAG | |

## Western blot analysis

Ovarian tissue and KGN cell lysates were extracted by RIPA lysis buffer (CWBIO, Beijing, China) and PMSF (Solarbio). The protein concentration was detected by a BCA Protein Assay Kit (CWBIO). Denatured proteins were separated by 10% SDS-PAGE, electrophoretically transferred onto polyvinylidene fluoride membranes (Meck Millipore, Merck & Co., Inc., NJ, USA). Then, transferred membranes were blocked by PBST (phosphate buffered saline [PBS] with 0.1% Tween-20) containing 5% skim milk for 2 h. The membranes were respectively incubated with antibodies against NF-κB p65 (1:1000 dilution, #8242, Cell Signaling Technology), IL-6 (1:1000 dilution, TD6087, Abmart Shanghai Co.,Ltd.), TNF-α (1:1000 dilution, PY19810, Abmart Shanghai Co.,Ltd.), TERT (1:1000 dilution, TD7129, Abmart Shanghai Co., Ltd.), Bax (1:1000 dilution, T40051, Abmart Shanghai Co.,Ltd.), Bcl-2 (1:1000 dilution, 13–8800, Thermo Fisher Scientific, Shanghai, China), Caspase-3 (1:1000 dilution, #9662, Cell Signaling Technology) and β-Tubulin (1:5000 dilution, R20005, Abmart Shanghai Co.,Ltd.) overnight at 4˚C, followed by HRP-conjugated affinipure goat anti-mouse IgG (H+L) (1:5000 dilution, SA00001-1; Protein Tech Group Inc., USA) or goat anti-rabbit IgG (H+L) (1:5000 dilution, SA00001-2; Protein T ech Group Inc.) for 2h at room temperature. Finally, eECL (CW0049M, CWBIO) was added, and the Tanon-5500 Chemiluminescence Imaging System was used to detect the chemiluminescence of protein bands. Finally, blotting images were quantified using Image J analysis software (JAVA image processing program, NIH, Bethesda, USA).

## Immunohistochemistry (IHC)

Samples of ovarian tissues was fixed in 4% paraformaldehyde overnight before being embedded in paraffin, and then were sequentially sectioned with 5 μM- slices. The endogenous peroxidase activity was blocked by 3% $H_2O_2$ for 30 min. The slices were put in the microwave in 0.1 M sodium citrate (pH 6.0) three times for antigen retrieval, permeabilized with 1% T riton X-100 and PBST for 30 min, and blocked with 5% bovine serum albumin for 45 min. Subsequently, the slides were incubated in a humidity chamber at 4˚C overnight with primary antibody NF-κB p65 (1:400 dilution, #8242; Cell Signaling Technology), IL-6 (1:100 dilution, TD6087, Abmart Shanghai Co.,Ltd.), TNF-α (1:200 dilution, PY19810, Abmart Shanghai Co., Ltd.), TERT (1:100 dilution, TD7129, Abmart Shanghai Co.,Ltd.), Bax (1:100 dilution, T40051, Abmart Shanghai Co.,Ltd.), Bcl-2 (1:200 dilution, 13–8800, Thermo Fisher Scientific, Shanghai, China), Caspase-3 (1:100 dilution, #9662, Cell Signaling Technology). After washing with PBST, the slices were incubated with Biotin-SP-conjugated Affinipure Rabbit Anti-Goat IgG (H+L) (1:200, SA00004–4; Protein Tech Group Inc.) at room temperature and 37˚C for 45 min. The slices were then incubated with HRP-conjugated Streptavidin (1:200, SA00001–0; Protein Tech Group Inc.) at 37˚C for 45 min. The slides were finally treated with 3,3-diaminobenzidine (DAB) (20× Metal Enhanced DAB Substrate Kit; Solarbio) to allow the brown staining of the positive cases. PBST was used as a negative control. Finally, the stained sections were observed and photographed under a light microscope (BX43, Olympus).

## Measurement of ovarian telomere length

Total RNA was extracted from ovarian tissue using TRIZOL reagent and reverse transcribed into cDNA. DNA was then used as a template for Rt-PCR analysis. The PCR reaction system consisted of 5 uL of 2×ChamQ SYBR qPCR Master Mix, 1 μL of primer mix, 1 uL of template DNA, and 3 uL of ddH2O. The reaction conditions for telomere gene Tel and reference gene 36B4 were basically the same, with 32 cycles for telomere gene and 40 cycles for reference gene, and melting curve analysis was performed after PCR reaction. Telomere length was

**Table 2. Primer sequences used for telomere length analysis.**

| Gene ID | | Sequence |
|---|---|---|
| Rat Telomere | Forward | GTAATTGCGTAAGACTTAAAACC |
| | Reverse | CCTAGAAATAAGAGGATTTAAACC |
| Rat 36B4 | Forward | ACTGGTCTAGGACCCGAGAAG |
| | Reverse | TCAATGGTGCCTCTGGAGATT |

analyzed by relative quantification method: Ct values of target gene and reference gene were obtained, $\Delta Ct = Ct^{Tel} - Ct^{36B4}$, telomere/single copy reference ratio (T/S) was calculated: $2^{-Ct} = (2^{CtTel}/2^{Ct36B4})^{-1}$, the relative T/S ratio $2^{-\Delta\Delta Ct} = 2^{-(\Delta Ct1 - \Delta Ct2)}$, this ratio corresponds to the relative telomere length of the samples, where $\Delta Ct1$ is the T/S of each sample and $\Delta Ct2$ is the T/S of the reference sample. The primer sequences used for amplification are shown in **Table 2**.

## Cell viability assay

Cell viability was measured by Cell Counting Kit-8 Solution Reagent (CCK-8, Beyotime, China). Briefly, cells were seeded in 96-well plates, and Wells without cells were set as blank control, positive control, and different concentrations of BAY 11-7082(0, 5, 10, 5, and 20 μm) and BIBR1532(0, 10, 20, 40, and 80 μM) for culture. Cell viability was examined by following standard procedures. Experiments were performed in triplicate.

## Cell culture and treatment

Human ovarian granulosa-like tumor cell line KGN, which originated from a stage Ⅲ invasive ovarian granulosa cell carcinoma in a 63-year-old woman, this cell line In review was considered as a model for understanding the regulation of steroidogenesis, cell growth, and apoptosis in human granulosa cells. In this study, KGN cells were purchased from Zhejiang Meisen Cell Technology Co., LTD (CTCC-003-0105). KGN cells were cultured with Dulbecco's Modified Eagle's Medium-high glucose (DMEM, Sigma, USA) supplemented with 10% Fetal Bovine Serum (FBS, Invitrogen 115 Gibco, USA) and maintained in an atmosphere of 5% $CO_2$ at 37˚C. KGN cells were plated in 6 cm plates at 106 per well. After starving for 24 h, the cells were treated without or with LPS (100 μg/mL) (I2643; Sigma, USA), BAY 11–7082 (10 μg/mL) (HY-13453, MCE, USA), and C(40 μg/mL) (HY-17353, MCE, USA) for 24 h.

## Flow cytometry was employed to detect cell apoptosis

KGN cells from drug-treated groups were digested using trypsin without EDTA (Biosharp, China). After terminating digestion, cell suspensions were collected in centrifuge tubes and centrifuged at 800 rpm for 5 minutes at room temperature, and the supernatant was discarded. The cells were then washed with pre-chilled 1×PBS (4˚C) and centrifuged at 800 rpm for 5 minutes. FITC-conjugated Annexin-V apoptosis detection kit (556547, Becton Dickinson, USA) was used to label KGN cells from each group for subsequent flow cytometry analysis.

## Statistical analysis

Data were analyzed using GraphPad Prism 8.0 (GraphPad Software, CA, USA) and presented as the mean ± standard deviation. Significant differences between/within groups were evaluated by the unpaired t-test or one-way ANOVA followed by Bonferroni 's post-hoc test. Correlation between variables was determined by Pearson's correlation coefficient. P value $<0.05$ was considered to be statistically significant.

## Results

### Letrozole and high-fat diet induced clinical PCOS-like changes in SD rats

As mentioned above, the pathogenesis of PCOS has been studied from many angles, but remains unexplored the role produced by the NF-κB-TERT pathway in the pathogenesis of PCOS, so we aims to elucidate this aspect in the current study. Therefore, a PCOS rat model was established by the treatment of HFD and letrozole. As shown in **Fig 1A and 1B**, the body weight and the ovarian weight of rats in the PCOS group was significantly increased than that of the control group. Compared with the control group, the serum levels of testosterone was significantly higher in PCOS group (**Fig 1C**).Estrous cycle disorder is one of the main characteristics of PCOS [38–40]. The normal estrous cycle in rats averages 4–5 days, which is generally divided into four stages—Proestrus, Estrus, Metestrus, and Diestrus [36]. As shown in **Fig 1D and 1E**, the control rats had a regular estrous cycle. Briefly, samples dominated by nucleated epithelial cells indicated the proestrus stage, and those with primarily cornified squamous epithelial cells indicated the estrus stage. The metestrus stage was indicated by mixed cells (nucleated, cornified, leucocytes), while the diestrus stage was dominated by leukocytes. Compared with control rats, the PCOS rats displayed disrupted estrous cycles, which comprised only the metestrus or diestrus phases. Appearance of representative ovaries. Compared with the control group, multiple follicles with cystic expansion presented vacuolated and disorganized structure (**Fig 1F**). Assessment of the ovarian morphology showed a significant decrease in the number of follicles and corpus luteum in PCOS rats compared with those in the controls, whereas antral follicles with cystic expansion were increased. The thickness of the granulosa cell layer was also reduced (**Fig 1G and 1H**).These results proved that we successfully constructed the PCOS rat model.

### Letrozole and high-fat diet induced PCOS increased the expression of NF-κB-related inflammatory factors and TERT in ovaries

Subsequently, to investigate the association between NF-κB-TERT feedback regulation and ovarian granulosa cell apoptosis in PCOS rats, we initially assessed the levels of NF-κB-related inflammatory factors in ovarian tissues from both groups to elucidate the link between chronic low-grade inflammation in the ovary and TERT expression. The qRT-PCR results demonstrated an upregulation of NF-κB, IL-6, TNF-α, and TERT in PCOS rats (**Fig 2A**). Western blot analysis further confirmed the elevated levels of NF-κB, IL-6, TNF-α, and TERT in the PCOS group compared to the control group (**Fig 2B and 2C**). Immunohistochemistry analysis also supported these findings by revealing increased expression of NF-κB, IL-6, TNF-α, and TERT in the PCOS group (**Fig 2D**). Our findings suggest that induction of PCOS through letrozole administration and a high-fat diet leads to enhanced expression of NF-κB-related inflammatory factors as well as TERT within the ovary. Additionally, telomere length was determined using qPCR analysis on ovarian granulosa cells from PCOS rats. utilized telomere-specific primers for this analysis. As depicted in **Fig 2E**, Compared with the control group, the telomere length of the cells in the ovaries increased by 1 to 2 times. These results indicate that the activity of inflammatory markers in ovarian cells of PCOS rats is increased and TERT function is increased.

### The increase of ovarian cell apoptosis in PCOS rats is positively correlated with NF-κb related inflammatory factors, TERT and apoptosis factors

To further validate our hypothesis, we proceeded to investigate the expression of apoptotic factors at both mRNA and protein levels. In comparison with the control group, we observed a

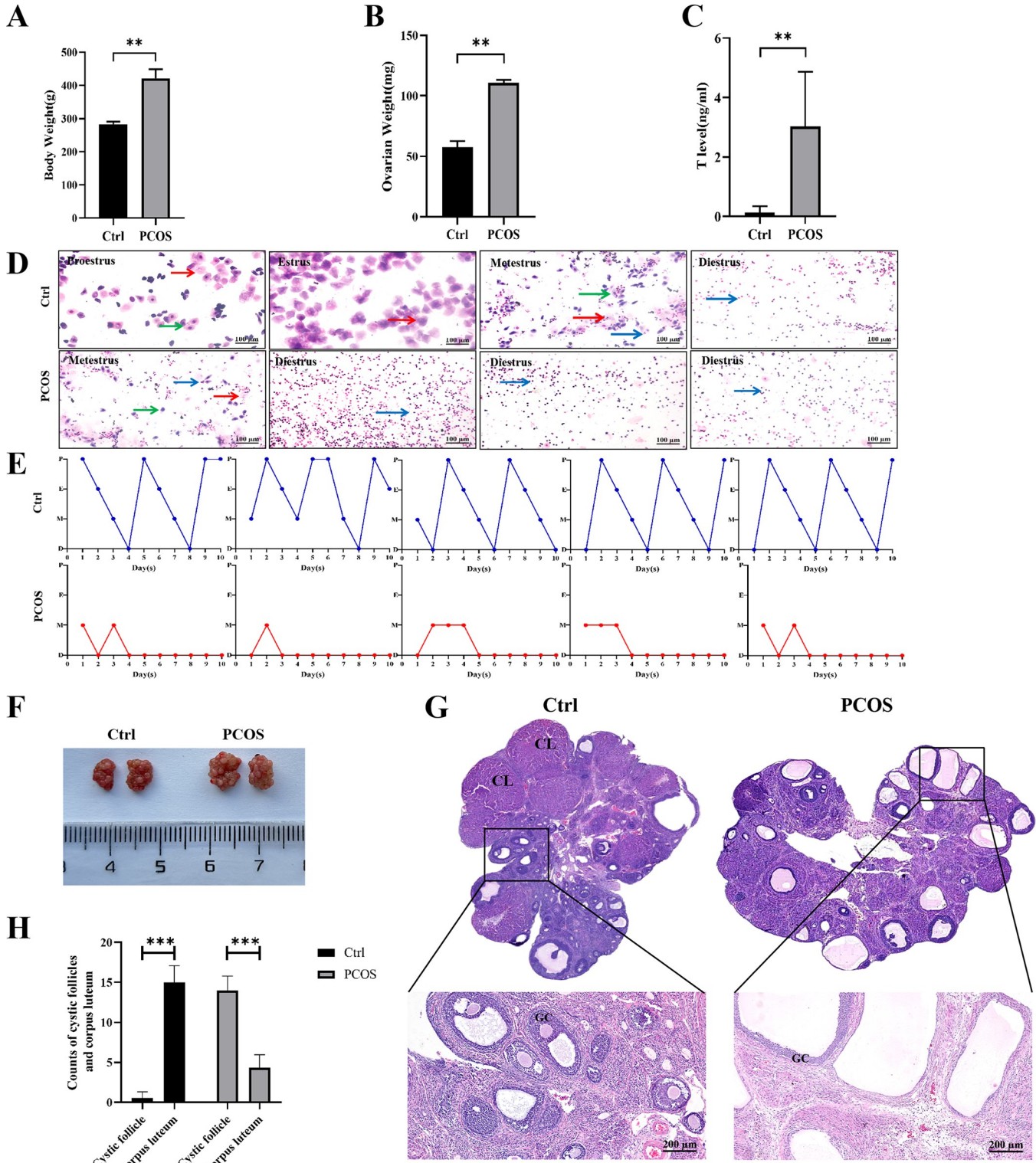

**Fig 1. Effects of letrozole and high-fat diet induction on body weight, ovarian weight, serum hormone levels, estrous cycle, and ovarian morphology.** A: Body weight of each group (n = 6 per group, day 60). B: Bilateral total ovarian weight in each group. C: Serum testosterone levels in each group (n = 6 per group). D: Cytological assessment of vaginal smears (n = 5 per group, day 51–60). E: Line chart of estrous cycle (n = 5 per group, day 51–60). F: Ovarian morphology of each group. G: HE staining scan and magnification of ovarian sections in each group (n = 6 per group, scale bar in upper is 500 μm). The lower panel show the higher magnification of the box area in the upper panel respectively. H: Counts of cystic follicles and corpus luteum (n = 6 per group). CL: corpus luteum; GC: granulosa cells. Significant differences between groups were indicated as **$P < 0.01$ and ***$P < 0.001$.

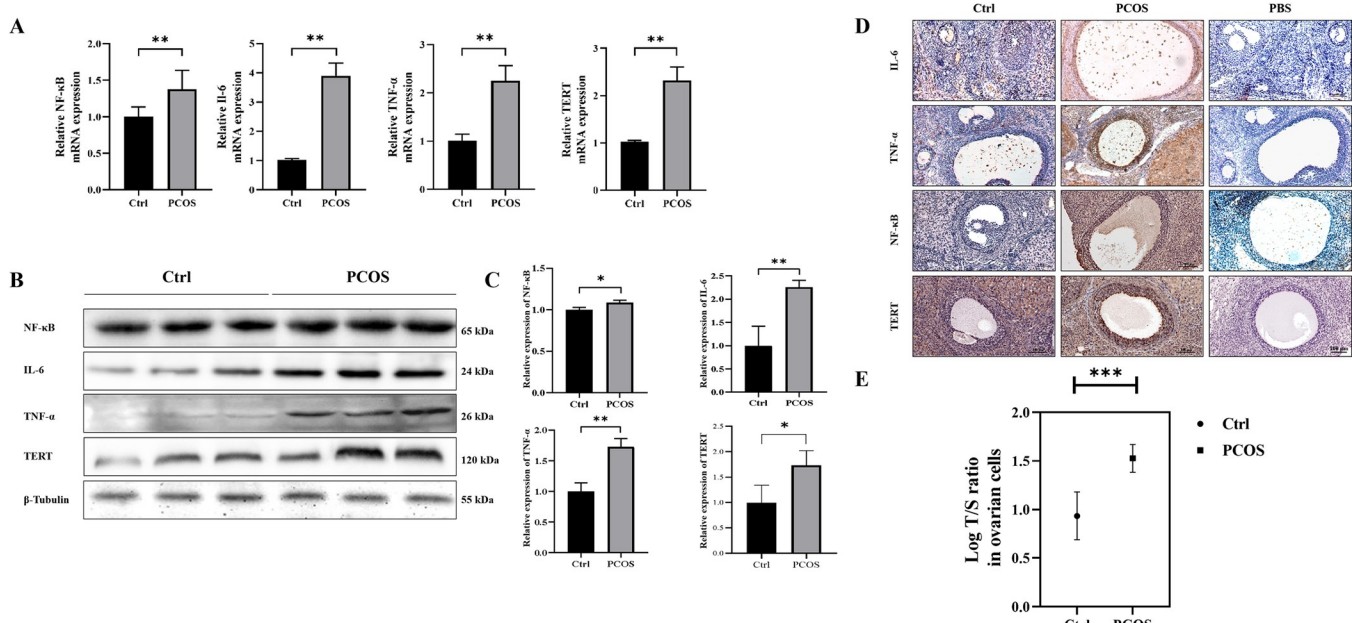

**Fig 2. The levels of NF-κB related inflammatory factors, TERT expression and telomerase relative length are significantly increased in the ovarian tissue of PCOS rats.** A: The expression changes of NF-κB, IL-6, TNF-α and TERT at mRNA level were detected by qRT-PCR (n = 3 per group). B-C: Western blotting was used to detect the protein expression of NF-κB, IL-6, TNF-α, and TERT, and ImageJ was used for quantitative analysis (n = 3 per group). D: Immunohistochemistry was used to detect the expression changes of NF-κB, IL-6, TNF-α and TERT at protein level (n = 4 per group), the lowest panel shows the negative control. E: Telomere length quantitative qRT-PCR results showed increased telomere length in the ovarian region of PCOS rats compared to controls (n = 3 per group). T/S ratio (telomere/single gene ratio); Data were presented as mean ± SEM. Significant differences were respectively presented as *$P < 0.05$, **$P < 0.01$, ***$P < 0.001$.

significant increase in the mRNA levels of Bax and caspase-3, along with a decrease in Bcl-2 mRNA level in GCs from the PCOS group (**Fig 3A**). Additionally, compared to the control group, GCs from the PCOS group exhibited elevated protein levels of Bax and caspase-3, while showing reduced levels of Bcl-2 (**Fig 3B and 3C**). Immunohistochemical staining analysis yielded consistent results (**Fig 3D**). Subsequently, we conducted an extensive analysis to explore any potential relationship between NF-κB-related inflammatory factors, TERT, and apoptotic factors in PCOS; aiming to elucidate whether chronic low-grade ovarian inflammation and up-regulation of TERT are associated with apoptosis. **Fig 3E** demonstrates a significant correlation among NF-κB-related inflammatory factors, TERT, and apoptotic factors. Specifically, Nf-κb-related inflammatory factors and TERT displayed positive correlations with pro-apoptotic markers such as Bax and Caspase-3; conversely exhibiting negative correlations with anti-apoptotic factor Bcl-2. These findings indicate that ovarian granulosa cells undergo apoptosis in PCOS rats, and underscore a positive correlation between NF-κB-related inflammatory factors, TERT, and apoptosis.

## After LPS stimulation, NF-κb-related inflammatory factors, hTERT expression and apoptosis of KGN cells increased, and there was a correlation among three

The results of animal experiments demonstrated a significant correlation between NF-κB, its downstream inflammatory factors, and TERT with apoptosis-related factors in PCOS rats. To further investigate the relationship between NF-κB-TERT regulation and PCOS granulosa cell apoptosis, KGN cells were treated with LPS at a concentration of 1μg/ml to establish an

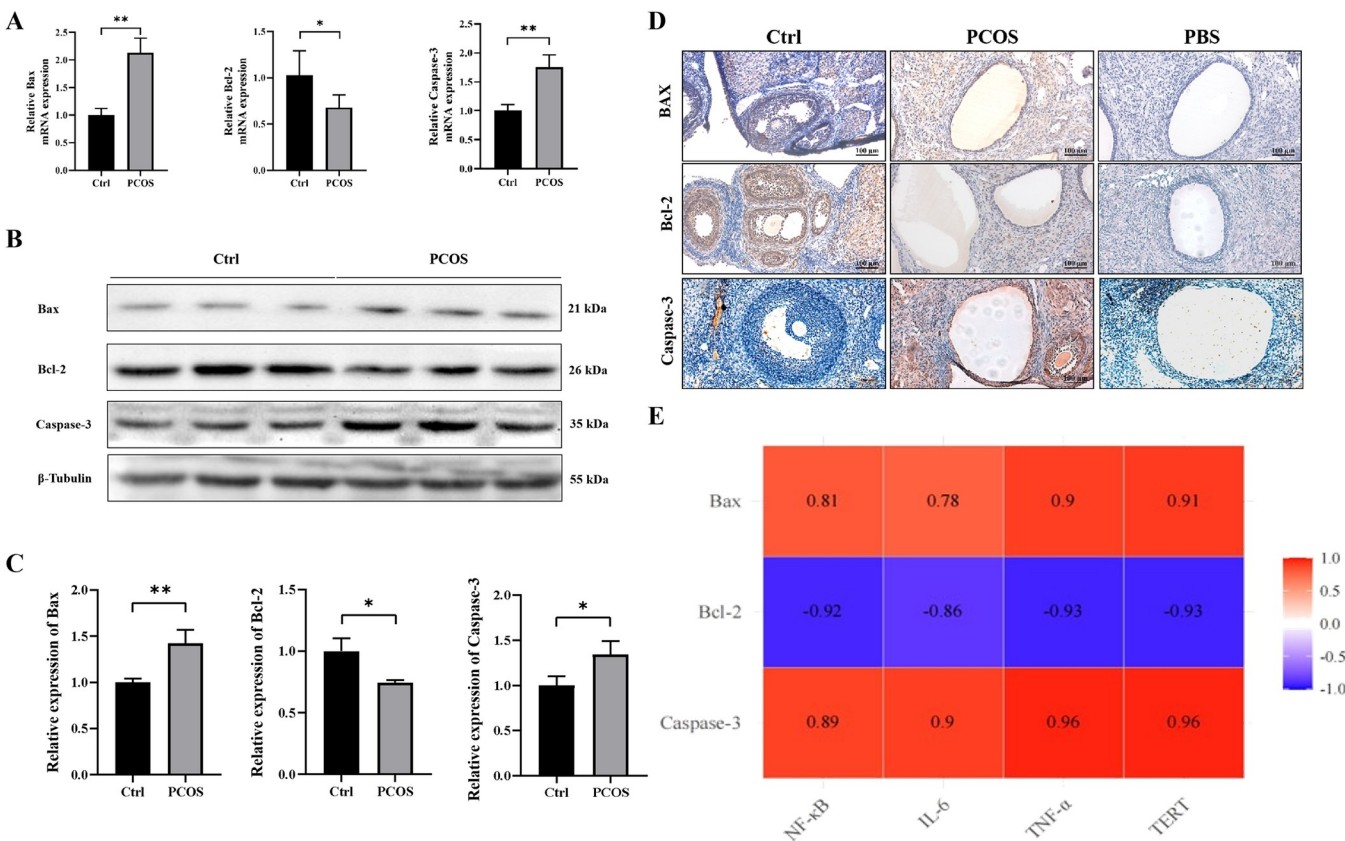

**Fig 3. Apoptosis is increased in the ovary of PCOS rats, and there is a correlation between NF-κB-related inflammatory factors, TERT and apoptosis factors.** A: The expression changes of Bax, Bcl-2 and Caspase-3 at the mRNA level were detected by qRT-PCR (n = 3 per group). B-C: Western blotting was used to detect the changes in the expression of Bax, Bcl-2 and caspase-3 at the protein level, and ImageJ was used for quantitative analysis (n = 3 per group). D: The expression of Bax, Bcl-2 and caspase-3 at the protein level was detected by immunohistochemistry (n = 4 per group), the lowest panel shows the negative control. E: Correlation analysis of inflammatory factors NF-κB, IL-6, TNF-α, telomerase reverse transcriptase TERT and apoptosis-related factors Bax, Bcl-2, Caspase-3 in ovarian tissue of rats in each group, Correlation between variables was determined by Pearson's correlation coefficient. $P < 0.05$ was considered to be statistically significant. Data were presented as mean ± SEM. Significant differences were respectively presented as *$P < 0.05$ and **$P < 0.01$.

inflammatory cell model. This model was divided into two groups: control group and LPS group. qPCR assay was employed to assess changes in mRNA expression levels of NF-κB, its downstream inflammatory factors, hTERT, and apoptosis-related factors in both groups. In the LPS group, there was a significant upregulation of mRNA expression for NF-κB p65 and its downstream inflammatory factors IL-6, TNF-α, hTERT as well as pro-apoptotic factors Bax and Caspase-3 (**Fig 4A**). Conversely, the expression of anti-apoptotic factor Bcl-2 was significantly downregulated. Based on our previous animal experiments which revealed a significant correlation between NF-κB, its downstream inflammatory factors, TERT with apoptosis-related factors in PCOS rat ovaries; we found that NF-κB along with its downstream inflammatory factors were positively correlated with pro-apoptotic markers while negatively correlated with anti-apoptotic markers. To validate these findings from animal experiments further analysis was conducted to explore the association between NF-kappa B along with its downstream inflammatory mediators (IL-6, TNF-α), hTERT and apoptotic markers within KGN cells. As depicted in **Fig 4B** it is evident that there exists a significant correlation among NF-kappa B along with its downstream inflammatory mediators (IL6, TNF-α) as well as hTERT concerning apoptotic markers. Specifically, NF-κB p65, IL-6, TNF-α, and hTERT were positively associated with pro-apoptotic proteins such as Bax and Caspase-3, while exhibiting a

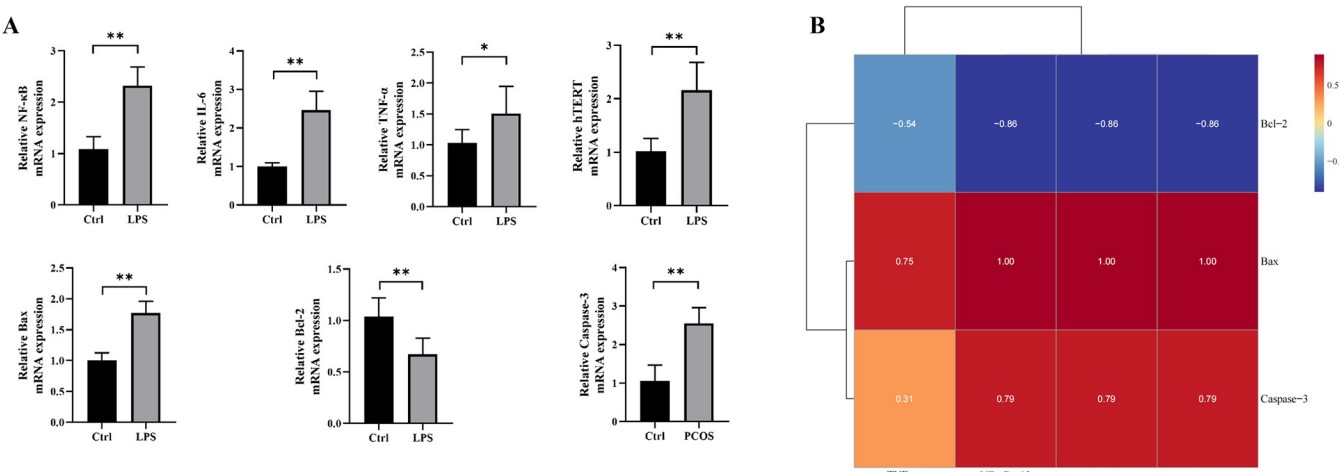

**Fig 4. LPS treatment increases the expression of NF-κB-related inflammatory factors, hTERT and cell apoptosis in KGN cell and correlation between NF-κB-related inflammatory factors, TERT and cell apoptosis factors after LPS stimulation in KGN cell.** A: The mRNA expression of NF-κB, IL-6, TNF-α, TERT, Bax, Bcl-2 and Caspase-3 in KGN cells in each group was detected by qRT-PCR. B: The correlation between inflammatory factors (NF-κB, IL-6, TNF-α, TERT) and apoptosis-related factors (Bax, Bcl-2, Caspase-3) in KGN cells after LPS stimulation was analyzed. Pearson correlation coefficient was used to determine the correlation between variables. P<0.05 was considered statistically significant. Data were presented as mean ± SEM. Significant differences were respectively presented as *P < 0.05 and **P < 0.01.

negative association with the anti-apoptotic protein Bcl-2. These results indicated that LPS-induced KGN cells exhibited increased expression of inflammatory factors and apoptosis, and NF-κB-related inflammatory factors and TERT were positively correlated with apoptosis.

## Inhibition of NF-κB and TERT reduced the expression of NF-κB-related inflammatory factors and TERT in KGN cells treated with LPS

To further elucidate the mechanism underlying NF-κB-TERT feedback regulation on apoptosis of ovarian granulosa cells in PCOS rats, we treated LPS-induced KGN cells with 10μg/ml BAY 11–7082 and 40 μg/ml BIBR1532, respectively. Experimental groups included Control (control), LPS (LPS induction only), LPS+NF-κB inhibitor (LPS induction + NF-κB inhibitor treatment), and LPS+TERT inhibitor (LPS induction + TERT inhibitor treatment). qPCR and Western Blot analyses were employed to assess the expression levels of NF-κB, its downstream inflammatory factors, and TERT in these four groups. Our results demonstrated a significant upregulation of NF-κB p65 along with its downstream inflammatory factors IL-6, TNF-α, and TERT in the LPS group compared to the control group. However, both the groups treated with NF-κB inhibitor BAY 11–7082 and TERT inhibitor BIBR1532 exhibited a significant downregulation of NF-κB p65 as well as its downstream inflammatory factors IL-6, TNF-α, and hTERT when compared to the LPS group (**Fig 5A and 5B**). To further investigate changes induced by LPS in KGN cells regarding inflammatory factors secretion, we quantified TNF-α and IL-6 levels in cell culture medium using ELISA assay (**Fig 5C**). Our findings revealed significantly increased levels of these inflammatory factors in the LPS group compared to both control and inhibitor groups.

## Inhibition of NF-κB and TERT reduced apoptosis in LPS-treated KGN cells

Upon observing a significant correlation among inflammatory factors, TERT, and apoptotic factors in KGN cells following LPS induction, we initiated further investigations into the alterations of apoptosis-related factors in these cells upon treatment with the NF-κB inhibitor BAY

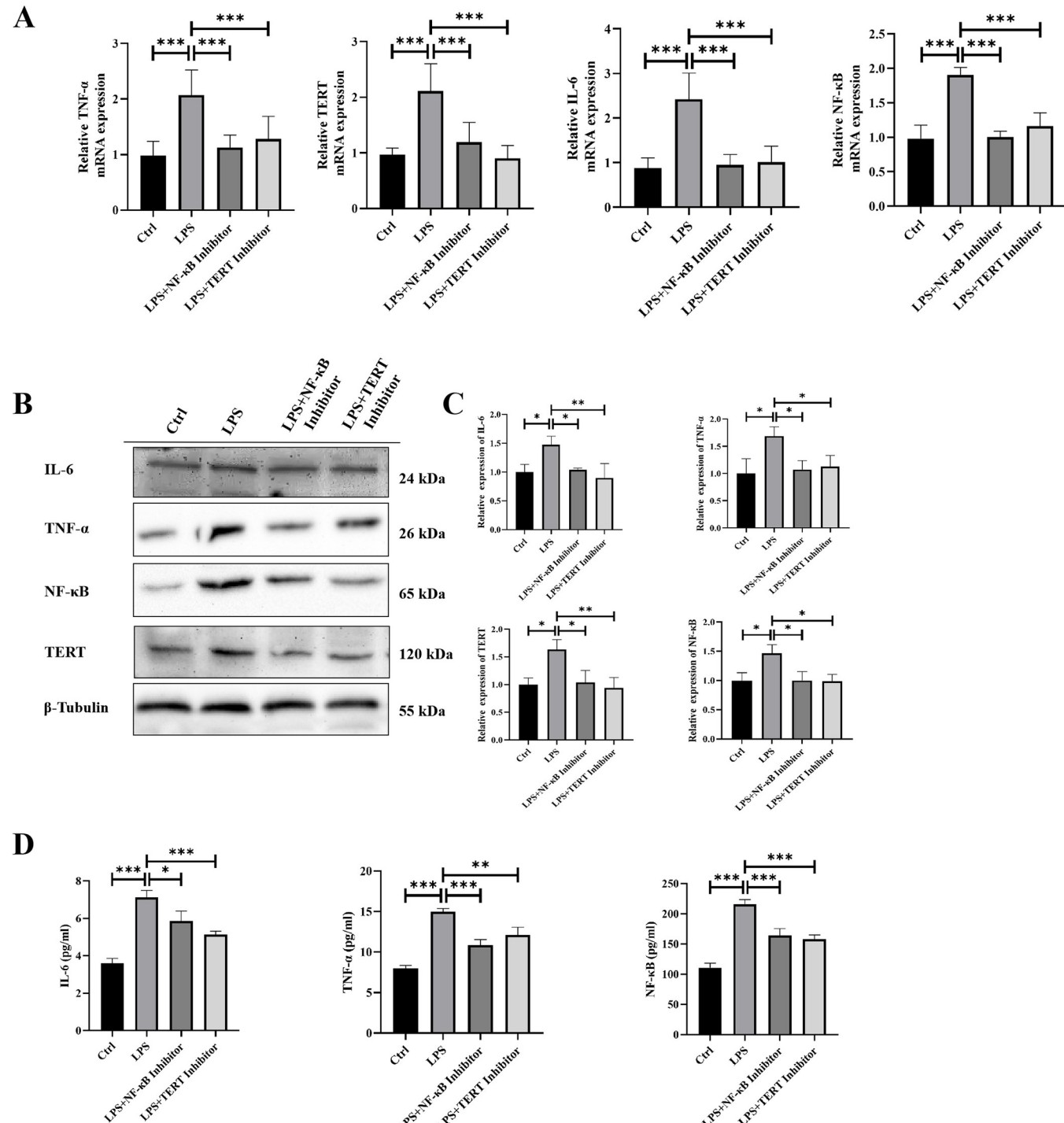

**Fig 5. Changes in the expression of NF-κB-related inflammatory factors, TERT and telomerase activity in KGN cells in each group.** A: The mRNA expression of NF-κB, IL-6, TNF-α and TERT in KGN cells of each group was detected by qRT-PCR. B-C: Western blotting detected the expression of NF-κB, IL-6 and TNF-α proteins in KGN cells of each group, and ImageJ was used for quantitative analysis. D: ELISA was used to detect the content of inflammatory factors (TNF-α, IL-6) in the cell culture medium. Data were presented as mean ± SEM. Significant differences were respectively presented as *$P < 0.05$, **$P < 0.01$, ***$P < 0.001$.

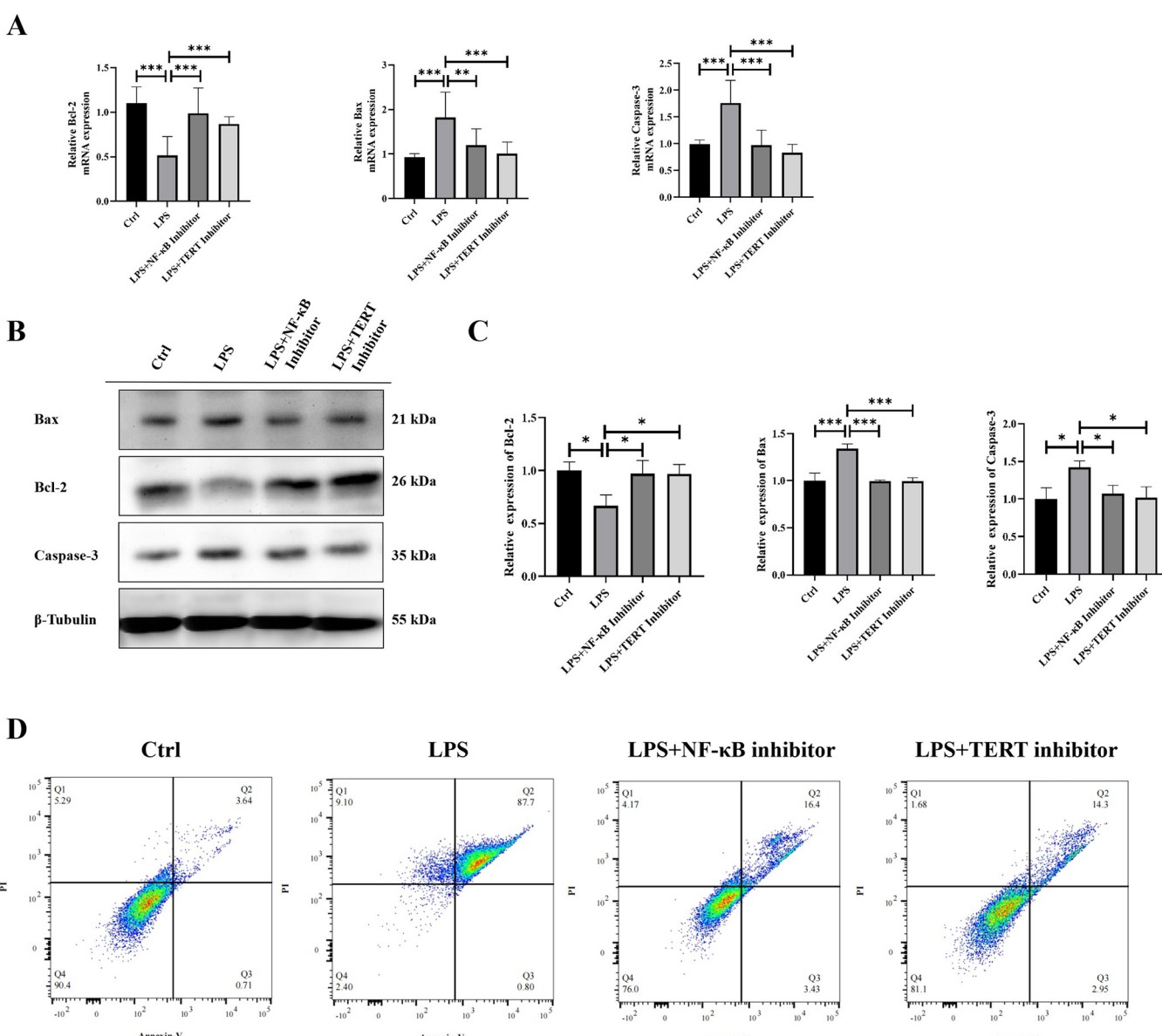

**Fig 6. Changes in the expression of apoptosis-related factors Bax, Bcl-2, and Caspase-3 in KGN cells in each group.** A: The mRNA expression of Bax, Bcl-2 and Caspase-3 in KGN cells of each group was detected by qRT-PCR. B-C: Western blotting detected the expression of Bax, Bcl-2 and Caspase-3 proteins in KGN cells of each group, and ImageJ was used for quantitative analysis. D: The apoptosis of KGN cells in each group was detected by flow cytometry. Data were presented as mean ± SEM. Significant differences were respectively presented as *$P < 0.05$ and ***$P < 0.001$.

11–7082 and the TERT inhibitor BIBR1532. The expression of apoptosis-related factors was assessed using qPCR and Western Blot analysis across four cell groups (**Fig 6A–6C**). Our findings revealed that compared to the control group, pro-apoptotic factors Bax and Caspase-3 were significantly up-regulated while anti-apoptotic factor Bcl-2 was significantly down-regulated in the LPS group. However, when compared to the LPS group, both NF-κB inhibitor BAY 11–7082 and TERT inhibitor BIBR1532 exhibited significant down-regulation of pro-apoptotic factors Bax and Caspase-3 along with a notable up-regulation of anti-apoptotic factor Bcl-2. Moreover, we employed Annexin V combined with propidium iodide labeled flow cytometry for apoptosis detection, followed by result analysis using FlowJo software.

According to the results depicted in **Fig 6D**, the apoptosis rate of KGN cells induced by LPS significantly increased compared to the control group. However, the apoptosis rate of LPS-induced KGN cells treated with NF-κB and TERT inhibitors approached that of the control group. These results suggest that chronic low-grade inflammation in PCOS patients triggers a positive feedback regulation of the NF-κB-TERT pathway, which subsequently leads to the release of downstream inflammatory factors IL-6 and TNF-α, thereby increasing the expression levels of pro-apoptotic markers BAX and Caspase-3. At the same time, it decreases the expression level of anti-apoptotic marker Bcl-2, which eventually leads to the apoptosis of granulosa cells.

## Discussion

PCOS is characterized by the presence of stunted follicles in the enlarged ovaries of affected females, typically accompanied by menstrual irregularities, hyperandrogenism, metabolic abnormalities, and mental disorders [41]. Currently, PCOS is the most common cause of female anovulation and infertility [42]. PCOS is a multifaceted systemic disease that significantly impacts women's overall health and well-being throughout their lifespan. It not only affects the reproductive endocrinology of young women, but also involves multiple systems such as glucose and lipid metabolism, cardiovascular and skin. However, the mechanism of polycystic ovary syndrome is still unclear, so it is urgent to clarify the exact pathogenesis of PCOS, improve the diagnostic criteria of PCOS, and develop more targeted drugs.

In recent years, the role of chronic low-grade inflammation in the pathogenesis of PCOS has attracted increasing attention. Recent studies have shown that inflammatory processes are closely related to ovulation and play an important role in ovarian follicle dynamics. A large body of evidence has shown that hyperandrogenism in PCOS patients is closely related to inflammation-related genes such as TNF-α, Tnf receptor 2 (TNFR2) and IL-6 [10]. Moreover, numerous studies have identified a strong association between PCOS-related inflammatory responses and visceral adipose tissue [43]. Abdominal obesity is common in PCOS patients. Visceral adipose tissue recruits immune cells by increasing the production and secretion of inflammatory factors monocyte chemoattractant proteins (MCP), leading to inflammatory response and maintaining the inflammatory state of adipocytes [44]. Hyperglycemia associated with insulin resistance in PCOS patients can also induce inflammatory responses, leading to NF-κb activation and oxidative stress. Oxidative stress is the key link of chronic low-grade inflammation in PCOS, and is closely related to the expression of related pro-inflammatory factors [44]. Contemporary biomedical research indicates that sustained activation of the inflammatory response in ovarian tissue plays a pivotal role in the reproductive endocrine dysfunction of PCOS [45]. On the one hand, the release of inflammatory cytokines in ovarian tissue directly damages granulosa cells, which is not conducive to the development of follicles, and then leads to rare ovulation or anovulation. On the other hand, it stimulates androgen production, which further aggravates ovulation disorder and leads to systemic hyperandrogenism. Simultaneously, the activation of the inflammatory response can interfere with insulin signal transduction, reducing insulin sensitivity and leading to insulin resistance and compensatory hyperinsulinemia. In conclusion, PCOS is closely linked to chronic low-grade inflammation.

As shown in the present study, we examined the changes in the expression levels of NF-κB and its downstream target genes IL-6 and TNF-α in ovarian granulosa cells of PCOS rats. The results showed that compared with the control group, the expression levels of NF-κB P65 and its downstream inflammatory factors IL-6 and TNF-α in the ovarian tissue of PCOS group were significantly increased. In order to further explore the chronic inflammatory response in

the granulosa cells of PCOS rats, we continued to construct the granulosa cell inflammation model by adding LPS, and continued to detect the expression changes of the above factors. The same results were found, the expression of the above factors in LPS group was significantly higher than that in control group. Molecular biological studies on PCOS suggest that the continuous activation of chronic microinflammation is one of the molecular mechanisms closely related to the occurrence and development of the disease [46].The above results verify that the ovarian granulosa cells of PCOS rats may have a chronic inflammatory response mediated by NF-κB.

Since both telomere length and telomerase activity are important factors affecting follicular development, it has been proposed that abnormalities in the telomere/telomerase system may be one of the pathogenic mechanisms of PCOS [23]. However, the existing research results based on the relationship between PCOS and telomere/telomerase system are still controversial. Wei [22] et al. found that the telomere length of granulosa cells in PCOS patients was significantly longer than that in non-PCOS patients undergoing in vitro fertilization (IVF), however, there was no statistically significant difference in the telomere length of peripheral blood leukocytes between the two groups. Changes in telomerase activity between the two groups were not reported Pedros et al. [23] measured telomere length and telomerase activity in various cell types in ovaries of PCOS patients compared to controls with normal menstrual cycles. Women in both groups underwent in vitro fertilization with intracytoplasmic sperm injection (IVF/ICSI). It was found that the telomerase activity of cumulus granulosa cells in PCOS group was significantly higher than that in control group. However, there was no significant difference in telomere length between the two groups. In addition, Wang et al. [21] conducted a study on women aged 20–30 years and compared 40 PCOS patients with 35 non-PCOS women and found that the telomere length of granulosa cells in the PCOS group was significantly longer than that in the control group, and there was no difference in the telomere length of peripheral blood leukocytes between the two groups. This is consistent with our study that telomere length and telomerase activity are significantly increased in PCOS. Whereas the persistent oxidative stress present in PCOS patients can lead to decreased telomere stability, and proinflammatory factors and ROS may lead to shorter telomere length in leukocytes [47] and reduced telomerase activity [48]. Li [49] et al. found that the telomere length of granulosa cells in PCOS patients was shorter than that in IVF patients without PCOS, but there was no significant difference in telomerase activity between the two groups. The influence might stem from insulin resistance, and the pivotal distinction between GCs and leukocytes lies in the fact that GCs possess telomerase activity, which leukocytes lack. In addition to oxidative stress, telomerase activity is another determinant of telomere length in GCs [22, 33]. Previous findings suggest that elevated T, a hallmark of PCOS, may increase telomerase activity [50]. Telomerase is a holoenzyme composed of two basic components: TERT and TERC. TERT is a protein with reverse transcriptase activity that synthetes a hexamer sequence at the end of the chromosome and is highly expressed in renewing tissues. It represents the main component of the holoenzyme, without which there is no enzyme activity [51]. TERT is a major determinant of telomerase activity, and its expression levels are consistent with telomerase activity. Studies have found that the TRET gene can regulate persistent chronic low-grade inflammation, which in turn can influence the transcription and expression of TERT-related genes, further strengthening the cycle of inflammatory signaling pathways [52]. The p38 MAPK signaling pathway plays an indispensable role in inflammation [53]. Research has shown that the p38 MAPK pathway can not only regulate the telomere/telomerase system by activating the NF-κB signaling pathway but can also cause changes in telomere length by regulating the expression levels of the shelterin protein complex components TRF1 and TRF2 [54, 55]. Concurrently, IL-6 and TNF-α can synergistically activate STAT3, promote

the physical association of STAT3 with NF-κB to form a complex, and activate the target gene hTERT. Due to the synergistic activation of STAT3, the transcription of TERT and telomerase activity increase with the activation of STAT3 and NF-κB [56]. Conversely, drugs that target the inhibition of the STAT3-TERT feedback loop can alleviate inflammation and slow disease progression [57]. Undoubtedly, the NF-κB signaling pathway is the most important pathway for regulating inflammatory responses [58]. In chronic inflammation, NF-κB has been found to exacerbate telomere dysfunction in mice [59]. Furthermore, activated NF-κB can increase the expression of TERT, and the addition of NF-κB inhibitors significantly downregulates the expression levels of TERT [60]. This indicates a positive correlation between NF-κB levels and telomerase activity. Moreover, IL-6 and TNF-α can upregulate TERT transcription and telomerase activity by activating and binding to NF-κB in macrophages [61]. This is consistent with our research findings. In this study, we investigated the expression of TERT, a key telomerase gene, in granulosa cells of PCOS rats. We found that the expression of TERT in the ovarian tissue of rats in the PCOS group was significantly upregulated compared with the control group. Therefore, we continued to verify at the cellular level, and we found that the expression of hTERT in granulosa cells was significantly up-regulated in the LPS group compared with the control group. Since LPS can induce the activation of NF-κB signaling pathway [62]. Combined with our previous results, we found that the expression of NF-κB and its downstream inflammatory factors and TERT was increased in ovarian granulosa cells of PCOS rats, and the NF-κB signaling pathway and hTERT expression were increased upon LPS induction. We further treated KGN cells with NF-κB and TERT inhibitors, and the expression of inflammatory factors and hTERT decreased. To confirm our idea, we went on to examine telomere length in ovarian cells from PCOS rats and found that telomere length was significantly longer in PCOS than in control rats. In this study, the telomere length of granulosa cells in PCOS rats was significantly longer than that in control rats, which may be due to the higher TERT activity in granulosa cells of PCOS rats. We speculate that the increased expression of TERT in ovarian granulosa cells of PCOS patients is related to the chronic inflammatory response mediated by NF-κB, and there is an interaction between the two.

Apoptosis is a physiological process of cellular clearance that is regulated by pro-apoptotic proteins belonging to the B-cell lymphoma (BCL) 2 family, BAX form transmembrane pores on the cell membrane, allowing the efflux of apoptotic factors [63]. Concurrently, the key apoptotic effector Caspase-3 can cleave and inactivate PARP, a factor involved in DNA repair [64]. Furthermore, NF-κB has been found to be activated during chronic low-grade inflammation and can control the downstream inflammatory cytokine TNF-α to mediate apoptosis. This may be a key factor leading to cellular apoptosis in chronic low-grade inflammation [65, 66]. Apoptosis plays a key role in follicular development. granulosa cells are one of the main functional cells involved in the development of follicles, and some studies have shown that a variety of clinical manifestations of PCOS patients are closely related to granulosa cell dysfunction [67]. Cataldo [68], Ding [69], Zheng [70] et al. found that the incidence of PCOS was closely related to the increase of apoptosis rate of ovarian granulosa cells. We examined the expression of Bax, Bcl-2 and Caspase-3 in the granulosa cells of PCOS rats. The results showed that Bax, Bcl-2 and Caspase-3 were expressed in the granulosa cells of PCOS rats. The expression of pro-apoptotic factors Bax and Caspase-3 in PCOS group was significantly up-regulated, and the expression of anti-apoptotic factor Bcl-2 was significantly down-regulated. Further experiments were carried out in KGN cells induced by LPS, The results showed that LPS induced significant up-regulation of Bax and Caspase-3 expression and down-regulation of Bcl-2 expression in granulosa cells. Furthermore, NF-κB inhibitor and TERT inhibitor were added to LPS-induced KGN cells, and the level of apoptosis was significantly decreased. By correlation analysis, the results showed that NF-κB and its downstream inflammatory factors

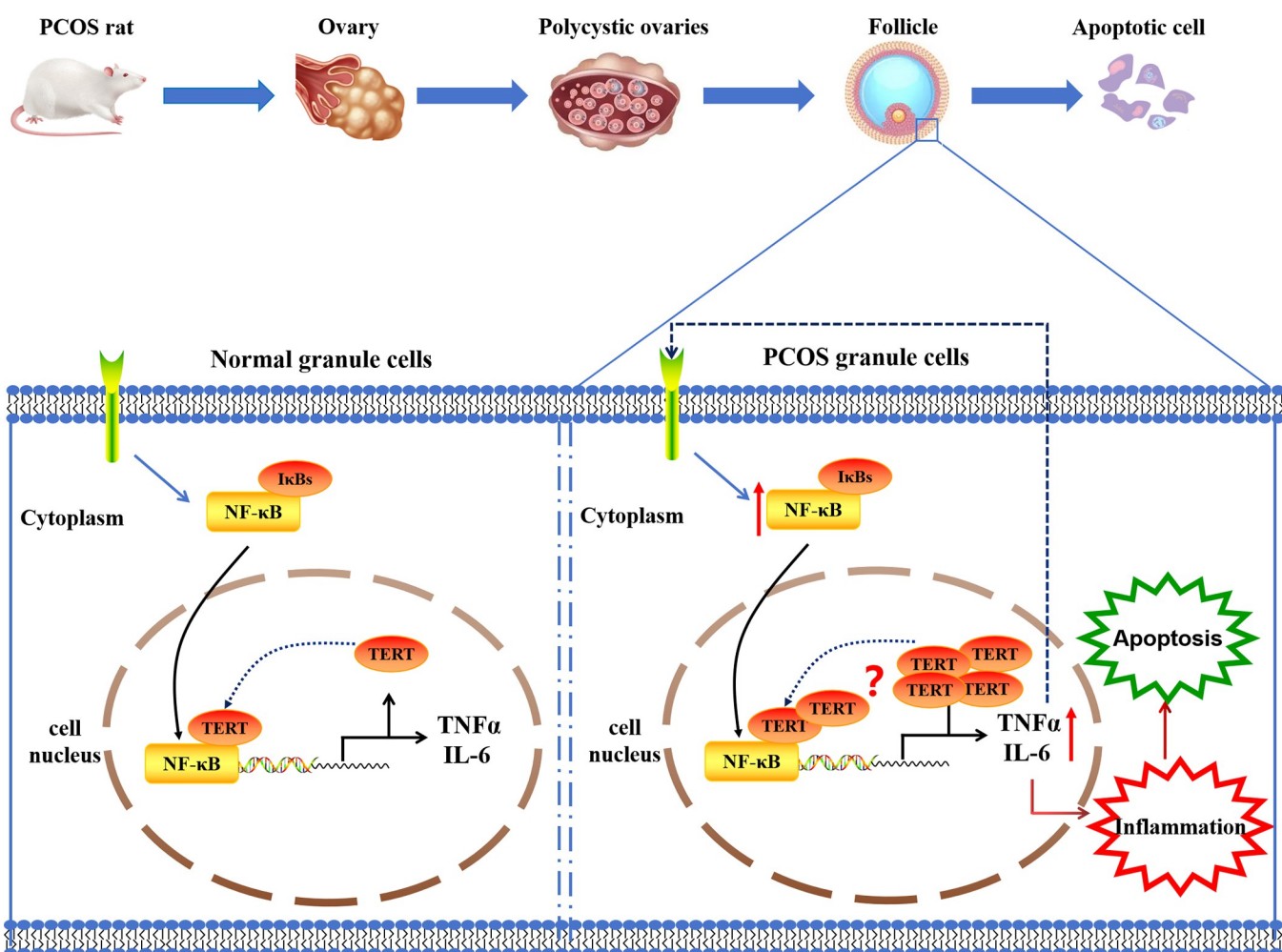

**Fig 7. The NF-κB-TERT in PCOS ovarian feedback regulation mechanism of simple model.** TERT may participate in regulating the NF-κB signaling pathway, activating downstream inflammatory factors IL-6 and TNF-α, mediating the chronic inflammatory response of PCOS, and inducing apoptosis of ovarian granulosa cells.

and TERT were significantly correlated with apoptosis related factors in the ovarian cells of PCOS rats. At the cellular level, we obtained the same results. The above experimental results verified that the ovarian granulosa cells of PCOS had apoptosis, which was also the key cause of PCOS, and NF-κB and its downstream inflammatory factors and TERT were significantly correlated with apoptosis related factors.

Targeted drugs that address the feedback mechanism between inflammation and telomere dysfunction may offer a novel therapeutic approach for PCOS in the future. Interventions that modulate and alleviate inflammatory responses have a positive impact on telomere dysfunction in PCOS patients. For instance, studies have found that Nicotinamide mononucleotide (NMN) has anti-inflammatory properties that can improve oocyte quality [71]. However, due to limitations in specimen collection, we did not collect follicular fluid from clinical PCOS patients to verify the involvement of TERT in the regulation of the NF-κB signaling pathway. Further research is required to confirm the interactive relationship between NF-κB and TERT in the follicular fluid of clinical PCOS patients (**Fig 7**).

## Conclusion

In conclusion, this study observed a significant correlation between NF-κB-TERT feedback loop and apoptosis-related factors in ovarian granulosa cells of PCOS rats. NF-κB-TERT exhibited positive correlations with pro-apoptotic factors and negative correlations with anti-apoptotic factors. TERT may participate in the regulation of NF-κB signaling pathway and activate the downstream inflammatory factors IL-6 and TNF-α to mediate the chronic inflammatory response of PCOS and induce the apoptosis of ovarian granulosa cells. Understanding the role of the NF-κB-TERT feedback loop in the pathogenesis and progression of PCOS can aid in elucidating the specific molecular mechanisms underlying this disease and provide novel insights for the development of targeted therapeutics.

## Supporting information

**S1 File. The original uncropped and unadjusted images underlying all blot or gel results.**
(RAR)

**S2 File. The original plotting data for Fig 7.**
(ZIP)

## Author Contributions

**Conceptualization:** Jiaming Zhang, Jing Zhang, Xiaocan Lei.

**Data curation:** Haoxuan Xue, Zecheng Hu.

**Funding acquisition:** Shun Zhang.

**Investigation:** Shun Liu.

**Methodology:** Haoxuan Xue, Zecheng Hu, Shun Liu, Wenqin Yang, Jiasi Li, Chulin Yan, Jing Zhang, Xiaocan Lei.

**Project administration:** Shun Zhang.

**Software:** Zecheng Hu.

**Supervision:** Wenqin Yang, Jiaming Zhang.

**Writing – original draft:** Haoxuan Xue.

**Writing – review & editing:** Haoxuan Xue, Jiaming Zhang, Jing Zhang, Xiaocan Lei.

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
