## [Decision Letter · Decision Letter 0]

20 Aug 2024

PONE-D-24-32182The mechanism of NF-κB-TERT feedback regulation of granulosa cell apoptosis in PCOS rats with follicular development disorderPLOS ONE

Dear Dr. Jiaming Zhang,

Thank you for submitting your manuscript to PLOS ONE. After careful consideration, we feel that it has merit but does not fully meet PLOS ONE’s publication criteria as it currently stands. Therefore, we invite you to submit a revised version of the manuscript that addresses the points raised during the review process.

The progression from describing the problem to presenting data and discussing the outcomes is logical and well-structured. However, the manuscript requires a revision of presentation. This includes enhancing the readability, improving the structure, and ensuring that technical terms and concepts are clearly defined. A proofreading is essential to eliminate any grammatical errors. Kindly check-up the manuscript and make the necessary changes as identified.

We look forward to receiving your revised manuscript.

Kind regards,

Akingbolabo Daniel Ogunlakin, Phd

Academic Editor

PLOS ONE

Journal Requirements:

"This work was supported by the Scientific Research Foundation of Hunan Provincial Education Department (No. 20A434), The Natural Science Foundation of Guangxi in China (No. 2020GXNSFAA297097), The Natural Science Foundation of Hunan Province, China (No. 2020JJ5511 and 2024JJ9373), Scientific Research Foundation of Hunan Provincial Education Department (No. 22B0406), Health Research Foundation of Hunan Provincial Health Commission Scientific Researchand (No. W20243058), Foundation of Hengyang City science and technology innovation plan (No. 202330046470) and the Hunan Province Innovation and Entrepreneurship Training Program for College Students (No. S202310555229, S202310555214 and D202305162026050688)."

4. In the online submission form, you indicated that [The datasets used and/or analysed during the current study are available from the corresponding author on reasonable request.]. 

7. PLOS requires an ORCID iD for the corresponding author in Editorial Manager on papers submitted after December 6th, 2016. Please ensure that you have an ORCID iD and that it is validated in Editorial Manager. To do this, go to ‘Update my Information’ (in the upper left-hand corner of the main menu), and click on the Fetch/Validate link next to the ORCID field. This will take you to the ORCID site and allow you to create a new iD or authenticate a pre-existing iD in Editorial Manager.

8. Your ethics statement should only appear in the Methods section of your manuscript. If your ethics statement is written in any section besides the Methods, please delete it from any other section. 

9. We note that Figure 7 in your submission contain copyrighted images. All PLOS content is published under the Creative Commons Attribution License (CC BY 4.0), which means that the manuscript, images, and Supporting Information files will be freely available online, and any third party is permitted to access, download, copy, distribute, and use these materials in any way, even commercially, with proper attribution. For more information, see our copyright guidelines: http://journals.plos.org/plosone/s/licenses-and-copyright.

a. You may seek permission from the original copyright holder of Figure 7 to publish the content specifically under the CC BY 4.0 license. 

Reviewers' comments:

Reviewer's Responses to Questions

**Comments to the Author**

1. Is the manuscript technically sound, and do the data support the conclusions?

Reviewer #1: Yes

Reviewer #2: Yes

2. Has the statistical analysis been performed appropriately and rigorously? 

Reviewer #1: Yes

Reviewer #2: Yes

3. Have the authors made all data underlying the findings in their manuscript fully available?

Reviewer #1: Yes

Reviewer #2: Yes

4. Is the manuscript presented in an intelligible fashion and written in standard English?

Reviewer #1: Yes

Reviewer #2: Yes

5. Review Comments to the Author

Reviewer #1: The study makes a valuable contribution to understanding the role of NFKB-TERT pathway in PCOS-related ovarian dysfunction. While the authors have conducted a thorough investigation, several considerations merit attention prior to the manuscript's acceptance.

Title: The title might feel a bit technical. Authors are encouraged to streamline the title.

Abstract:

- Technical terms could be avoided.

- Brief context should be added.

- Lack of quantitative data in the abstract.

Introduction:

- Authors are encouraged to remove redundancies, and improve transition/flowing.

- Apoptosis molecular mechanism should be expanded.

- Clearly articulate the research hypothesis.

Methodology:

- Authors must mention randomization/blinding during allocation of rats into groups.

- The study uses a well-established method to induce PCOS. It would be beneficial to add a reference.

- Were all the rats equally affected? How did the researchers ensure consistency across the PCOS group.

- Authors should mention whether the hormone levels were assessed at specific times in the oestrous cycle.

- Western blot: how was the normalization performed (which protein)? Indicate the statistical method.

- IHC: indicate the quantification method.

- Cell viability assay: specify the positive and the negative control. Replace the “ctrl” by “control”: this applies on the whole document.

Discussion:

- Lack of clarity in linking telomere length and PCOS.

- The inconsistent reporting of telomerase activity is not efficiently explained.

- While NFKB is undoubtedly important, the discussion should consider other signalling pathways (MAPK, JAK/STAT..).

- Need for detailed exploration of the specific apoptotic pathway involving Bax, bcl-2, and caspase-3. How does the pathways interact with the inflammatory signalling?

- Authors are encouraged to integrate the clinical relevance of the findings (new diagnostic criteria or targeted therapy….).

- Improve the flow between the topics.

- How does targeting theses pathways could enhance the therapeutic strategies? Discuss the potential therapies.

- Address the limitations of the study.

Conclusion: Conclusion could be improved. Figure 7 could be transferred to the end of the discussion.

Reviewer #2: The progression from describing the problem to presenting data and discussing the outcomes is logical and well-structured.

However, the manuscript requires a revision of presentation. This includes enhancing the readability, improving the structure, and ensuring that technical terms and concepts are clearly defined. A proofreading is essential to eliminate any grammatical errors.

Kindly check up the the manuscript and make the necessary changes as identified.

6. PLOS authors have the option to publish the peer review history of their article (what does this mean?). If published, this will include your full peer review and any attached files.

Reviewer #1: **Yes: **Amel Elbasyouni

Reviewer #2: **Yes: **Thomas Abu

---

## [Author Response · Author response to Decision Letter 0]

4 Sep 2024

Dear editors and reviewers,

Thank you very much for reviewing our manuscript titled “The mechanism of NF-κB-TERT feedback regulation of granulosa cell apoptosis in PCOS rats with follicular development disorder.” (PONE-D-24-32182) and for giving us an opportunity to revise the manuscript. We also greatly appreciate the reviewers for their critical review of the manuscript with thoughtful and constructive comments.

Academic Editor

Respond to these comments:

1.Please ensure that your manuscript meets PLOS ONE's style requirements, including those for file naming.

Thank you very much for your suggestions. Following the editor's recommendations, we have revised the manuscript to meet the style requirements of PLOS ONE.

2.To comply with PLOS ONE submissions requirements, in your Methods section, please provide additional information regarding the experiments involving animals and ensure you have included details on (1) methods of sacrifice, (2) methods of anesthesia and/or analgesia, and (3) efforts to alleviate suffering.

Thank you very much for the editor's suggestions. We have included additional information regarding the animal experiments in the 'Methods' section to comply with PLOS ONE submission requirements. This includes details on the methods of sacrifice, anesthesia, and efforts to alleviate suffering.

3.Please provide an amended statement that declares *all* the funding or sources of support (whether external or internal to your organization) received during this study, please also include the statement “There was no additional external funding received for this study.” in your updated Funding Statement.

We sincerely appreciate your feedback. Following the editor's suggestions, we have included a revised funding statement in the Cover letter and have added the statement "There was no additional external funding received for this study." in the Funding Statement section.

4.In the online submission form, you indicated that [The datasets used and/or analysed during the current study are available from the corresponding author on reasonable request.].

Thank you very much for your suggestions. In compliance with the requirements of PLOS ONE, we have made revisions to ensure that all relevant data can be accessed within the paper and its supporting information files.

5.When completing the data availability statement of the submission form, you indicated that you will make your data available on acceptance. We strongly recommend all authors decide on a data sharing plan before acceptance, as the process can be lengthy and hold up publication timelines. Please note that, though access restrictions are acceptable now, your entire data will need to be made freely accessible if your manuscript is accepted for publication. This policy applies to all data except where public deposition would breach compliance with the protocol approved by your research ethics board. If you are unable to adhere to our open data policy, please kindly revise your statement to explain your reasoning and we will seek the editor's input on an exemption. Please be assured that, once you have provided your new statement, the assessment of your exemption will not hold up the peer review process.

Thank you very much for the editor's reminder. In accordance with PLOS ONE's guidelines, we fully comply with the PLOS ONE open data policy.

6.PLOS ONE now requires that authors provide the original uncropped and unadjusted images underlying all blot or gel results reported in a submission’s figures or Supporting Information files.

Thank you very much for the editor's reminder. In accordance with the requirements of PLOS ONE, we have provided all original uncropped and unadjusted images underlying all blot results in the Supporting Information section.

7.PLOS requires an ORCID iD for the corresponding author in Editorial Manager on papers submitted after December 6th, 2016. Please ensure that you have an ORCID iD and that it is validated in Editorial Manager. 

Thank you very much for your reminder. We have now linked the corresponding author's ORCID iD.

Your ethics statement should only appear in the Methods section of your manuscript. If your ethics statement is written in any section besides the Methods, please delete it from any other section.

Thank you very much for the editor's reminder. We have removed any ethical statements that were written in sections other than the Methods section.

We note that Figure 7 in your submission contain copyrighted images. All PLOS content is published under the Creative Commons Attribution License (CC BY 4.0), which means that the manuscript, images, and Supporting Information files will be freely available online, and any third party is permitted to access, download, copy, distribute, and use these materials in any way, even commercially, with proper attribution.

Thank you very much for the editor's reminder. We confirm that the mechanism figure in Figure 7 was conceived and drawn by us without any copyright disputes.

Please review your reference list to ensure that it is complete and correct. If you have cited papers that have been retracted, please include the rationale for doing so in the manuscript text, or remove these references and replace them with relevant current references. 

We sincerely appreciate the editor's reminder. We have carefully reviewed our reference list and found that we did not cite any retracted articles.

Reviewer #1

The study makes a valuable contribution to understanding the role of NFKB-TERT pathway in PCOS-related ovarian dysfunction. While the authors have conducted a thorough investigation, several considerations merit attention prior to the manuscript's acceptance.

Respond to these comments:

1. The title might feel a bit technical. Authors are encouraged to streamline the title.

Thank you very much for the suggestions provided, which pointed out the need to reduce some professional expressions in the title to more comprehensively summarize the main theme of the article. Following the reviewer's advice, we have changed the title from "The mechanism of NF-κB-TERT feedback regulation of granulosa cell apoptosis in PCOS rats with follicular development disorder." to "The mechanism of NF-κB-TERT feedback regulation of granulosa cell apoptosis in PCOS rats."

2. Technical terms could be avoided.

Thank you very much for your suggestion. We have avoided the use of technical jargon in the abstract. We have made the correction on page 2, lines 45-47, and added the full term for the acronym LPS upon its first appearance.

3. Brief context should be added.

Thank you very much for your constructive feedback. We acknowledge that the abstract was lacking some introduction to the main topic. Following the reviewer's suggestion, we have added a 'Brief context' in the abstract section on page 2, lines 32-36, where we provide a concise overview of the relationship between inflammation and TERT in PCOS.

4. Lack of quantitative data in the abstract.

Thank you very much for your meticulous suggestions. We have enhanced the results section of the abstract by including quantitative data related to the findings, such as the magnitude of the P-values.

5. Authors are encouraged to remove redundancies, and improve transition/flowing.

Thank you very much for your feedback highlighting the need for better transitions and flow in the introduction section of our manuscript. Following your advice, we have revised the introduction to ensure a more coherent and logical progression of ideas.

6. Apoptosis molecular mechanism should be expanded.

Thank you very much for the reviewer's suggestion to include molecular mechanisms of apoptosis in the introduction, which will make the article more comprehensive. Following your advice, we have revised the introduction on pages 4, lines 99-104, and 107-109, to expand on the molecular mechanisms of apoptosis.

7. Clearly articulate the research hypothesis.

Thank you very much for the reviewer's insightful suggestion. As recommended, we have refined our research hypothesis in the introduction. We have revised the statement from "It is still not clear whether TERT can mediate chronic inflammatory response and induce granulosa cell apoptosis by participating in the regulation of NF-κB signaling pathway." to "Hence, TERT may mediate chronic inflammatory response and induce apoptosis of granulosa cells by participating in the regulation of the NF-κB signaling pathway." This change provides greater clarity and aligns with the findings of our study.

8. Authors must mention randomization/blinding during allocation of rats into groups.

hank you very much for the reviewer's careful reading, and we sincerely apologize for our oversight. Following the reviewer's suggestion, we have amended the text to "the rats were randomly divided into the control group (n = 6) and the PCOS group (n = 6)."

9. The study uses a well-established method to induce PCOS. It would be beneficial to add a reference.

We appreciate the important suggestions from the reviewer. To enhance the persuasiveness of the article, we have added references at the point where the PCOS model is induced.[1]

10. Were all the rats equally affected? How did the researchers ensure consistency across the PCOS group.

We greatly appreciate your valuable feedback. Yes, all rats were subjected to the same treatment. We ensured the consistency of the PCOS group by analyzing the estrous cycle changes through vaginal smears, the morphology of follicles through HE staining, and the changes in relevant serum hormones in the PCOS rats.

11. Authors should mention whether the hormone levels were assessed at specific times in the oestrous cycle.

We have conducted assessments of the estrous cycle and serum hormones following the methodologies outlined in the works of Zuo, Ling et al., and Ji, Rui et al., which involves the detection of serum hormones after the euthanasia of the rats. We are grateful for your valuable suggestions and plan to assess hormone levels at specific times during the estrous cycle in our future model construction.[2, 3]

12. Western blot: how was the normalization performed (which protein)? Indicate the statistical method.

Thank you very much for your valuable suggestion. We have used β-Tubulin protein for normalization. The statistical method we employed is one-way ANOVA, which has been described in the 'Statistical Analysis' section of the Materials and Methods.

13. IHC: indicate the quantification method.

We are very grateful for your reminder. Our immunohistochemistry experiments were conducted based on the methodologies from the articles of Yu, Jin et al., and Xu, Kaili et al..[4, 5] But we did not perform quantification for them. In our study, we only preliminarily explored the localization of NF-κB and its downstream inflammatory factors, apoptosis-related factors, and TERT in the ovaries. Thank you again for your important reminder; we have learned that ImageJ can be used for counting positive cells and plan to apply it in future experiments.

14. Cell viability assay: specify the positive and the negative control. Replace the “ctrl” by “control”: this applies on the whole document.

Thank you very much for your reminder. Following the reviewer's suggestion, we have revised the "Cell viability assay" section to read: "Cells were seeded in 96-well plates, with wells without cells serving as blank controls, positive controls, and different concentrations of BAY 11-7082 (0, 5, 10, and 20 μM) and BIBR1532 (0, 10, 20, 40, and 80 μM) for culture." We apologize once again for our negligence and have replaced "ctrl" with "control."

14. Lack of clarity in linking telomere length and PCOS.

Thank you very much for the suggestions provided by the reviewer. Following the reviewer's advice, we have made revisions in the discussion section. We have more clearly articulated the link between telomere length and PCOS. Studies have indicated that the telomere length of granulosa cells in PCOS patients is significantly longer than in non-PCOS patients, which is consistent with our research findings.[6]

15. The inconsistent reporting of telomerase activity is not efficiently explained.

Thank you very much for your suggestions. Although some studies have indicated that pro-inflammatory cytokines and reactive oxygen species (ROS) may lead to a decrease in telomerase activity in leukocytes of PCOS patients[7]. Our study's focus is primarily on the granulosa cells in PCOS, not leukocytes. Research has suggested that this difference may arise because granulosa cells, as actively dividing reproductive cells, have telomerase activity, which leukocytes lack[6]. Moreover, the PCOS model in our study is characterized by high testosterone levels and persistent low-grade inflammation. Previous studies have indicated that elevated testosterone, a signature marker of PCOS, may increase the activity of telomerase[8]. In response to the reviewer's feedback, we have made revisions in the discussion section to reflect these key points.

16. While NFKB is undoubtedly important, the discussion should consider other signalling pathways (MAPK, JAK/STAT..).

Thank you very much for the comments raised by the reviewer. Discussing other relevant signaling pathways will undoubtedly make our article clearer. In accordance with the reviewer's suggestions, we have made revisions on pages 16, lines 459 to 477, in the discussion section, where we further elaborated on the MAPK-TERT and STAT3-TERT feedback pathways.

17. Need for detailed exploration of the specific apoptotic pathway involving Bax, bcl-2, and caspase-3. How does the pathways interact with the inflammatory signalling?

We would like to express our gratitude for your meticulous and rigorous review. Following your suggestion, we have further discussed the apoptotic mechanisms of Bax, bcl-2, and Caspase-3, as well as how they interact with inflammatory cytokines, in the section on page 17, lines 496 to 503.

18. Authors are encouraged to integrate the clinical relevance of the findings (new diagnostic criteria or targeted therapy….).

Thank you very much for the valuable feedback from the reviewer. Relating the research findings to clinical applications can certainly enhance the readability and practical value of this paper. Following the reviewer's suggestions, we have made revisions on page 18, lines 522 to 523, where we propose new potential diagnostic targets.

19. Improve the flow between the topics.

We sincerely appreciate the valuable feedback provided by the reviewer. In accordance with the reviewer's suggestions, we have polished the manuscript to enhance the flow between topics.

20. How does targeting theses pathways could enhance the therapeutic strategies? Discuss the potential therapies.

We appreciate the suggestions made by the reviewer. Discussing potential therapeutic approaches would greatly enhance the clinical application value of this paper. Following the reviewer's advice, we have made revisions on page 18, lines 523 to 526.

21. Address the limitations of the study.

Thank you for the precious suggestions provided by the reviewer. Due to the limitations in obtaining specimens, we did not explore the regulatory relationship between TERT and NF-κB-related inflammatory factors in the follicular fluid of clinical PCOS patients. We are grateful once again for the reviewer's reminder, as this is an important part of our upcoming research. Following the reviewer's advice, we have made revisions on page 18, lines 527 to 530, where we discuss the limitations of this study.

21. Conclusion could be improved. Figure 7 could be transferred to the end of the discussion.

Thank you for the valuable suggestions provided by the reviewer. In accordance with the reviewer's comments, we have revised and enhanced the conclusion section. Additionally, we have moved Figure 7 to the end of the discussion section.

REFERENCES

1. Liang, A., et al., Resveratrol Improves Follicular Development of PCOS Rats by Regulating the Glycolytic Pathway. Molecular Nutrition & Food Research, 2021. 65(24).

2. Zuo, L., et al., Therapeutic potential of icariin in rats with letrozole and high-fat diet-induced polycystic ovary syndrome. Eur J Pharmacol, 2023. 953: p. 175825.

3. Ji, R., et al., Carnosol inhibi

---

## [Editor Report · Decision Letter 1]

16 Sep 2024

The mechanism of NF-κB-TERT feedback regulation of granulosa cell apoptosis in PCOS rats

PONE-D-24-32182R1

Dear Dr. Jiaming Zhang,

We’re pleased to inform you that your manuscript has been judged scientifically suitable for publication and will be formally accepted for publication once it meets all outstanding technical requirements.

Kind regards,

Akingbolabo Daniel Ogunlakin, Phd

Academic Editor

PLOS ONE
---

## [Editor Report · Acceptance letter]

16 Oct 2024

PONE-D-24-32182R1 

PLOS ONE

Dear Dr. Zhang, 

I'm pleased to inform you that your manuscript has been deemed suitable for publication in PLOS ONE. Congratulations! Your manuscript is now being handed over to our production team.

Kind regards, 

on behalf of

Dr. Akingbolabo Daniel Ogunlakin 

Academic Editor

PLOS ONE